# PoC (Proof of Concept) for Performance Monitoring Platform of Container Terminals

**Nam Kyu Park [1,\*] and Jung Hun Lee [2]**

[1]   Department of International Logistics, Tongmyong University, Busan 48520, Korea
[2]   Busan Development Institute, 955, Jungang-dearo, Busanjin-gu, Busan 47210, Korea; ljh4139@bdi.re.kr
\*   Correspondence: nkpark@tu.ac.kr; Tel.: +82-010-3575-1004

**Abstract:** The purpose of this study is to prove the concept of the performance monitoring system of container terminals. PoC (Proof of Concept) is a realization of a certain method or idea in order to demonstrate its feasibility. The port authorities, such as government or local authority, are continually checking the performance of the terminals they invested in and want to reflect it in the development policy. They also want to increase competitiveness by checking performance levels, such as port handling volume, calling ships, resource utilization, and congestion. PPI (Port performance indicators) are classified into four categories: output (production), productivity, utilization, and service. In this study, 15 monitoring indicators for each stage by dividing the process from the ship's entry to departure are defined. Four indicators, such as ship waiting ratio at anchorage, berth occupancy, storage occupancy, and truck turnaround time, are selected as PoC of monitoring platform. In addition, a method of collecting, processing, and expressing data on these four indicators in real time is presented. There are three steps to create PPI on monitoring platform. The information required for PPI is to be collected from the Port-MIS (Management Information System) and TOS (Terminal Operating System) databases. Second, the collected data from external entities are stored into the database after verification and classification. Third, descriptive PPI, predictive PPI are generated based on the input data. This study provides a 4-tier framework from the conceptual platform with the key elements of data presentation, data process and data interface and middleware. As a result of the study, it is proved to select monitoring indicators, define external entities, define internal elements of the system, develop systems, and present indicator results. However, in the process of collecting data outside the system, we have found there is confidential data of individual terminals. To this end, it is important to establish a mutual cooperation system for data collection.

**Keywords:** container terminal operation; port performance indicator; port monitoring platform

---

## 1. Introduction

The characteristics of smart port is that the cycle of decision making is shortened by going through the stages of data collection, analyzing, diagnosis, prediction and prescription [1]. The port authorities have invested a large amount of investment to construct and operate a container terminal, but it has not provided the monitoring platform to evaluate the performance of the terminal in the perspective of Port Authority. This paper is to prove whether the monitoring platform is useful for performance improvement by constructing a prototype [2] of port performance monitoring system.

PoC (Proof of Concept) is a realization of a certain method or idea in order to demonstrate its feasibility [3]. Port Authority including central or local government is continually checking the performance of the terminal they invested in and want to reflect it in the development policy. They also want to increase competitiveness by checking performance levels, such as cargo volume, calling ships, resource utilization, and congestion.

In order to immediately check the terminal performance, data must be input from individual terminals in real time. This study start defining the monitoring indicators for each stage by dividing the process from the ship's entry to departure. We will develop a prototype of performance monitoring platform to conceptually prove that it works as intended.

As monitoring system is to be used by Ministry of Ocean and Fishery (MOF), Port Authority (PA) and TOC (Terminal Operating Company) [4], it is necessary to analysis the relationship between source data as external entities and the system as internal system. The process, ERA (Entity Relationship Attribute) diagram and User View for creating PPI (Port Performance Indicator) must be designed during implementing PoC.

The definition and calculation formula of PPI has been studied by researchers and practitioners, but there is no systematic approach to creating them for Port Authority in real time as a platform type. The contribution of this paper is that the existing PPI was systematically classified and a new PPI, such as truck turnaround time, was proposed. Furthermore, it proved a system that can produce the defined PPI in real time by linking with the existing Terminal Operating System (TOS) and Port Management Information System (MIS).

Based on PPI, the paper discusses what data to collect, how to build a database, what type of system architecture does it have as a tool of PoC. Source data is collected from MOF and TOC and the output of monitoring platform is provided to MOF, PA, TOC, shipping company, etc. The data used in this study are batch data and have limitations in real-time data monitoring. This paper consists of six steps as follows.

(1) First step is to define a PPI to understand the container terminal's performance. PPI is defined according to the criteria of output, productivity, utilization and service.

(2) Second step is to collect data from TOC. This is the data regarding to berth facility and activity, yard facility and activity during one year (refer to Appendix A Tables A1–A4). For data collection. There are 19 terminals for collecting data within Busan port, Incheon port, Gwangyang port, Pyeongtaek port, and Ulsan port.

(3) Third step is to collect data from Port-MIS of MOF. Data consists of port code, port name, call sign, number of arrivals by year, ship name, facility code, facility name, port facility use purpose code, port facility use purpose name, start date and time, end of use date and time (refer Appendix A Tables A5 and A6).

(4) Fourth step is to design the architecture of the monitoring platform. In this study, four components: data presentation, data process and data interface and middleware are designed.

(5) Fifth step is to design and implement database. Then, the collected data is input into the database in order to produce the result.

(6) Sixth step is to design the PPIs' presentation format which is suitable for performance management as user view.

(7) Seventh step is to create a program that connects the database and graphic tool, SQL (Structured Query Language), web program language.

## 2. Methods

To develop a PPI monitoring platform, the development methodology, such as waterfall, spiral, agile rapid development, software prototyping, and incremental, is required [5]. In this study, rapid prototyping will be used for monitoring platform development because prototyping can improve the quality of requirements and specifications provided to developers [6]. This refers to the creation of a model that will eventually be discarded rather than becoming part of the final delivered software. After preliminary requirements gathering is accomplished, a simple working model of the system is constructed to visually show the users what their requirements may look like when they are implemented into a finished system [7]. The prototype is developed through the following procedure (refer Table 1).

**Table 1.** Research steps and results.

| Research Steps. | Description | Findings |
|---|---|---|
| PPI (Port Performance Indicator) Definition | Selected 14 PPIs for monitoring that PA (Port Authority) are interested in | Figures 1 and 2 Table 3 |
| User Query Definition | User queries are defined using PPI | Table 3 |
| Data Collection | Selecting 19 terminals for collecting data, including Busan Port | Appendix A Table A1 Table 4 |
| Data Item and Format Design | Design of data items related to terminal specifications, ship entry and exit, cargo handling, utilization of equipment, and truck entry and exit records | Appendix A Tables A1–A5 |
| Design of Business Flow Diagram | Defines external entities and internal components of the system. | Figure 3 |
| Design of System Architecture | Design system internal components consisting of data presentation, data process and data interface and middleware. | Figure 4 |
| Entity Relationship Diagram Design | Design ERD (Entity Relationship Diagram) for 4 representative PPIs in the ship entry and departure process. | Figure 5, 7, 9, and 11 |
| Programming for Prototype | Four PPIs were programmed using a database, graphic tool, and web programming language. | Figure 6, 8, 10 and 12 |

(1) First is to identify basic requirements: Determine basic requirements including the input and output information desired. Details, such as security, can typically be ignored. In this step, the PPIs to be monitored are defined. PPI selection is defined by referring to previous studies and the procedure of call ship and cargo in port. After selection of PPI, they are confirmed through consultation with officials of PA.

(2) Second is to develop initial prototype: After the initial prototype is developed that includes only user interfaces and database interface. In this step, the PPI defined in step (1) is created. Here, proof of concept is attempted for four of the PPIs. The four PPIs are representative of ship entry and departure processes, and consist of ship waiting ratio, berth occupancy, storage utilization, and truck turnaround time.

(3) Third is to review the prototype: The customers, including end-users, examine the prototype and provide feedback on potential additions or changes.

(4) Fourth is to revise and enhance the prototype: Using the feedback both the specifications and the prototype can be improved. Negotiation about what is within the scope of the contract/product may be necessary. If changes are introduced, then a repeat of steps (3) and (4) may be needed. The result is shown on confidential web address [8].

## 3. Results

### 3.1. PPI for Ship Arrival—Unloading—Stocking—Ship Depart

To identify the PPI for monitoring ship activity, the ship's arrival-berthing-loading/unloading-departure process should be described in Figure 1. When the ships arrive at port, the berth availability is to be identified before berthing. Especially, the berth scheduling should be checked in advance to ensure the availability of the berth. In order to alleviate potential congestion issues and avoid the cargo delivery delays to the end customer, TOC should improve the effectiveness of the terminal operations by monitoring PPI [9]. TOC strive to achieve rapid unloading and loading as ships become larger container vessels, which corresponds to a reduction of the time in port for the vessels [10]. To make

ships at Yangsan Port congested and shorten unloading time, Double-Line Shipping Mooring (DLSM) mode in enforced [11]. For monitoring port capacity, BAP (Berth Allocation Problem) must be solved in extensive simulations, needed to account for ship traffic and handling times uncertainties [12]. Efficient seaside operations are critical for TOC performance, as disruptions in the seaside operations may significantly delay service of the arriving vessels [13]. It is an important issue to measure the performance of the berth and prepare countermeasures due to the enlargement of ships and the increase in cargo volume. The solution affects the operational performance of the whole terminal container [14]. If all berths are occupied by ships, then a ship has to wait at anchorage. When the berth is empty, the pilot will embark for berthing, i.e., POB (pilot on board). After the ship is alongside at berth, loading and unloading can be started. If the cargo handling is complete, the ship departs the port. In the ship arrival and berthing process, the indicators of ship waiting time, ship waiting rate and ship turnaround time can be generated in the category of service. The time of the ship's anchoring and POB are recorded on Port-MIS which is managed by MOF. The berthing and departing time are recorded on TOS which is managed by TOC. Specially, ship waiting time is recorded in PFU (Port Facility Usage) entity in PORT-MIS, which is used when a ship requires a specific berth. Besides, ships' service time, the number of containers handled and the number of equipment used are recorded in TOS (Terminal Operating System).

·      Ship waiting time is defined as total hours that vessels wait for a berth [15].
·      Ship waiting rate is defined as ship waiting time divided by total service time at berth [15].
·      Ship turnaround time is defined as total time spent by a ship in port [15–17].

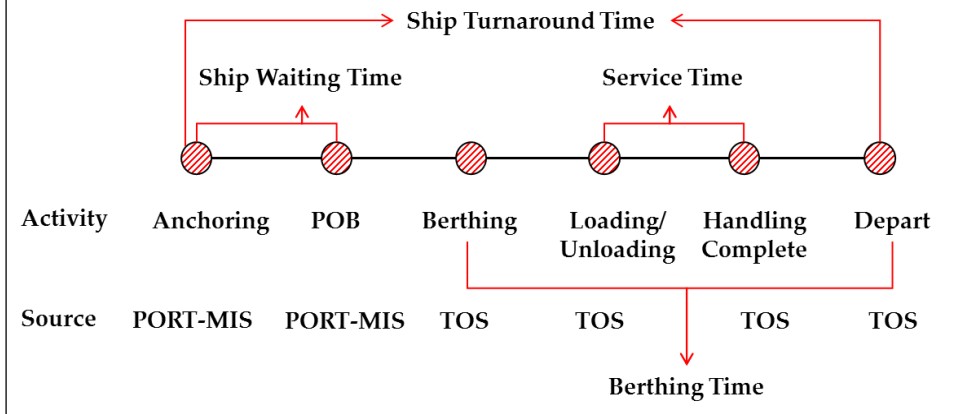

**Figure 1.** Procedure of ship arrival-waiting-berthing-handling-departure.

### 3.2. Ship Berthing-Unloading-Ship Departing Process

When the ship docks, Quay Cranes (QCs) are used for unloading and loading. When the loading and unloading work is completed, the ship departs the berth. In this process, performance indicators of quay crane productivity, ship productivity, berth productivity, berth throughput, ship throughput, and berth occupancy are derived with following definition.

·      Quay crane productivity is defined as handled containers divided by total number of crane-hours worked [15–18].
·      Ship productivity is defined as handled containers divided by ship-hours worked at berth [2,15–17].
·      Berth productivity is defined as handled containers divided by service-hours worked at berth [16–18].
·      Berth throughput is defined as handled containers at berth in a time of period [16,18].
·      Ship throughput is defined as handled containers at berth and in port in a time of period [15,16].
·      Berth occupancy is defined as the ratio of time that the berth is occupied by a vessel to the total time available in that period [15,16,18].

### 3.3. PPI for Truck Gate-In and Gate-Out Process

When the truck arrives at the gate, it checks congestion in the yard to determine whether to wait. The time of gate-in is recorded on CODECO (Container gate-in/gate-out report message) of UN/EDIFACT (United Nations for Electronic Data Interchange for Administration, Commerce and Transport) in TOS. After the truck load and unload containers, it leaves the gate of which time is recorded on CODECO in TOS. In this process, performance indicators of gate utilization, truck turnaround time and truck waiting time are derived with following definition

·   Gate utilization is defined as the ratio of time that the gate is occupied by trucks to the total time available in a time period [17,18].
·   Truck turnaround time is defined as total time spent by a truck in terminal [17,18].
·   Truck waiting time is defined as total times that trucks wait for stocking [17].

### 3.4. PPI for Stocking on Yard-Handling-Leaving Process

After unloading, the YT (Yard Tractor) or AGV (Automated Guided Vehicle) transfers the container from the ship to the storage site. The containers stay at storage during a dwell time, then leaves storage by external truck (refer Figure 2). In this process, performance indicators of yard throughput, yard productivity, yard utilization, container dwell time, and equipment utilization are derived with following definition.

·   Yard throughput is defined as total containers that handled in the yard areas [18].
·   Yard productivity is defined as yard throughput divided by total area of yard [18].
·   Yard utilization is defined as containers on hand divided by total available slots [16,18].
·   Container dwell time is defined as the period containers stay at the terminal [16,17].
·   Equipment utilization is defined as the proportion of time that it was effectively deployed over a specified period [17,18].

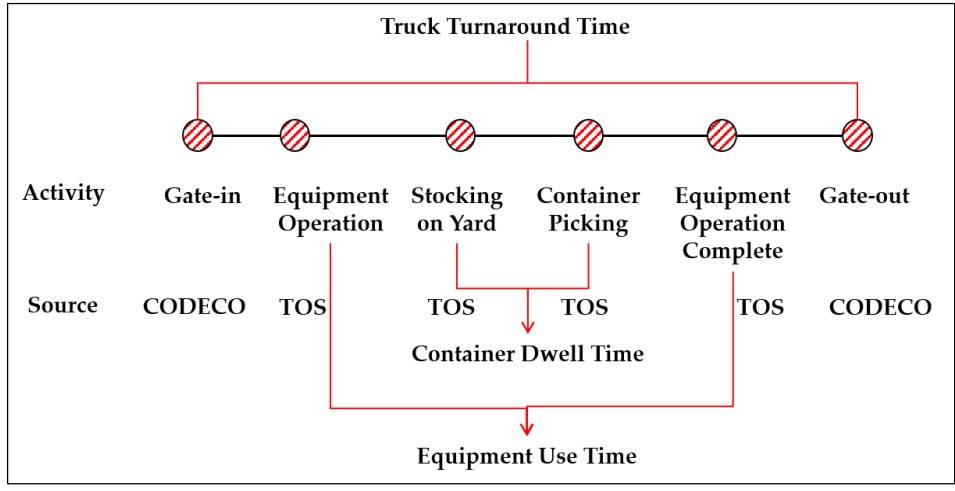

**Figure 2.** Procedure of truck gain-in-picking-stocking-delivery at storage.

### 3.5. Classification and Semantic Analysis of PPI

Thomas and Monie (2002) classified PPIs as production i.e., output, productivity, utilization, and service measure [18]. Production indicators are measured ship throughput, berth throughput, yard throughput as the level of the business activity. Productivity indicators are measured ship productivity, berth productivity, crane productivity and yard productivity as the ratio of output to input. Utilization indicators refer to how intensively the terminal resources are used. They are measured berth occupancy, yard utilization, gate utilization, equipment utilization. Service indicators refer to

customer satisfaction with terminal services to customers. They include ship turnaround time, road vehicle turnaround time. According to the same classification, World Bank [15], Hebel Mwasengark (2012) [16] defines PPI as shown in Table 2. Amr Arisha and Amr Mahfouz (2009) summarized researchers' papers in Table 2 (Peter.B Marlow et al., 2003 [19]; Wayne Talley, 2006 [20]; K. Dahal, 2003 [21]; Hugh S., 2000 [22]; and Ani Dasgupta et al., 2000 [23]).

**Table 2.** Port performance indicators (PPI) by precedent studies.

| Category | Thomas and Monie | World Bank | Hebel Mwasenga | Amr Arisha and Amr Mahfouz |
|---|---|---|---|---|
| Output (Production) | Ship Throughput, Berth Throughput, Yard Throughput, | - | Ship Throughput Berth Throughput | - |
| Productivity | Ship Productivity Berth Productivity Crane Productivity Yard Productivity | Ship Productivity Berth Productivity Crane Productivity | Ship Productivity Berth Productivity Crane Productivity Gang Productivity | Tons per ship-hour in port Tons per gang hour Number of cargo handled per resource (crane, labor) Handling rate of discharge operation |
| Utilization | Berth Occupancy, Yard Utilization, Gate Utilization, Equipment Utilization | Berth Occupancy | Berth Occupancy Yard Utilization. | Resource Utilization (Quay, Storage, Gate, Equipment), Ship's capacity utilization |
| Service | Ship Turnaround Time Truck Turnaround Time | Ship Turnaround Time, Ship Waiting Rate, Container Dwell Time | Ship Turnaround Time Truck Turnaround Time Container Dwell Time Equipment Availability | Truck Waiting Time |

## 4. Results

### 4.1. User View of Monitoring Platform

User view means a view of part or all of the contents of a database specified to facilitate a particular purpose or user activity. It is a partial and/or redefined description of the logical schema of the database [22]. The PPI definition is based on the user view of the monitoring system. The user view requires close communication with the user. In this study, it was defined through an interview with the official of Port Policy Department of MOF.

The user views were designed as a way to show a single or multiple PPIs in one screen. The point is the time period of the data is to be considered. The retrieval cycle is weekly, a monthly or annual basis. The second is to enable future prediction by presenting the time series trend equation of PPI (refer Table 3).

**Table 3.** PPI and user query in period.

| Category | PPI | User Query in Period |
|---|---|---|
| Output (Production) | · Calling Ship and Ship Throughput · Calling Ship and Berth Throughput · Berth Throughput and Yard Throughput | · What is container volume by full, empty, refrigerator, oversize and dangerous type? · What is calling ships of mother and feeder of the terminal? · How does the container volume change when the calling ships increases? · How does the throughput at storage change when the berth throughput increases? |

**Table 3.** *Cont.*

| Category | PPI | User Query in Period |
|---|---|---|
| Productivity | · Ship Productivity<br>· Crane Productivity<br>· Gang Shift Productivity<br>· Terminal Area Productivity | · What is GBP and NBP of the terminal?<br>· What is crane productivity of the terminal?<br>· What is gang shift productivity of the terminal?<br>· What is terminal area productivity of the terminal?<br>· How does GBP increase when terminal throughput increases?<br>· How does GBP increase when calling ships increases? |
| Utilization | · Berth Occupancy<br>· Yard utilization<br>· Equipment Utilization | · What is the berth occupancy of the terminal?<br>· What is yard utilization of the terminal?<br>· What is equipment utilization of the terminal?<br>· How does the occupancy of the berth change when the berth throughput increases?<br>· How does the occupancy of the berth change when the number of incoming ships increases?<br>· How does yard utilization change when berth occupancy increase? |
| Service | · Ship Waiting Rate<br>· Truck Turnaround time<br>· Container Turnaround time<br>· Crane Intensity<br>· Container Dwell Time | · What is the waiting time, service time and the number of waiting ship rate of the terminal?<br>· What is the truck turnaround time of the terminal?<br>· What is the container turnaround time of the terminal?<br>· What is crane intensity of the terminal?<br>· What is container dwell time by full, empty, refrigerator, oversize and dangerous type? |

*4.2. Collecting Input Data for PoC of Platform*

From Port-MIS, a total of 150,000 basic information was collected with 63,333 records of "ship entry and departure information" and 87,464 records of "port facility usage" during 2016–2018. Among these, "ship entry and departure information" records the ship's arrival and departure time, but has limitations that the shipping company's initial report information is not continuously updated. Therefore, in this study, it was decided that "ship entry and departure information" would not be used for PPI platform. Fifty-six million one hundred ninety-eight thousand one hundred eighty-three records of CODECO (Container gate-in/gate-out report message) were collected (refer Table 4).

From the TOS, raw data, such as dimension of quay wall and equipment, equipment usage status, ship entry and exit information, and storage information, were collected. Analyzing the collected information, there is a gap in the level of TOS, showing a lot of difference in the quality of information collected. Some container terminals do not manage the ship's call sign, so there is a problem in handling the connection with Port-MIS information of the Ministry of Oceans and Fisheries. This problem can be overcome in using the bypass method of tracking the call sign in reverse by using the information on "ship entry and departure information" and "port facility usage" of PORT-MIS. Basically the standard TOS and standardization of code is necessary to solve the problem. The information received from the

container terminal was collected in the form of Excel, with a total of 87,948 records, with 69,490 ship entry and departure information and 18,458 records including facility, yard utilization, equipment utilization, and truck arrival and departure.

**Table 4.** Collect data from Port Management Information System (MIS) and Terminal Operating System (TOS).

| Source | Name of Table | Number of Records |
|---|---|---|
| Port MIS | Whip entry and departure information | 63,333 |
| | port facility usage | 87,464 |
| | CODECO (Container gate-in/gate-out report message) | 56,198,183 |
| TOS | Ship entry and departure information | 69,490 |
| | Facility, Yard utilization, Equipment utilization, Truck activity in yard | 18,458 |

### 4.3. Business Flow Diagram

There are three steps to create PPI on monitoring platform. The information required for PPI is to be collected from the Port-MIS and TOS databases. External entities consist of PORT-MIS, TOS, and survey data of TOC, including domestic and overseas terminal. Second, the collected data from external entities are stored into the database after verification and classification. Third, descriptive PPI, predictive PPI are generated based on the input data (refer Figure 3). Additionally, diagnosis, prescription analysis will be useful to improve performance of container terminal (Evans [24]; Lustig et al., [1]; Davenport and Harris [25]).

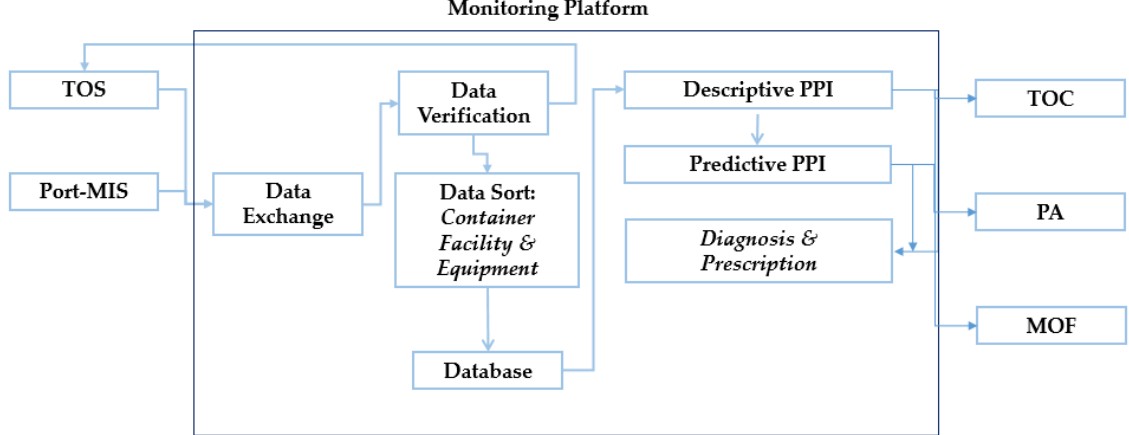

**Figure 3.** Business flow diagram.

After data transmitted from outside is divided into container data, and facility data, they stored in the database. Database design begins with ERA (Entity Relationship Attribute) diagram. Entities are shown as boxes in the ERA diagram and have an entity name; usually names are required to be unique. Attributes are generally shown as annotations of the entity boxes. Relationships are shown as lines between entity boxes [26,27].

If the ship's activity at the calling port is expressed in ERA diagram, it can be composed of a facility usage entity and a ship's berthing entity. As the primary keys of ship berthing entity and facility usage entity has different attributes, this problem is solved by finding and linking attributes with the same meaning.

*4.4. System Architecture of Monitoring Platform*

In this section, a conceptual framework for POI information infrastructure for container terminal will be provided. The proposed infrastructure enables the online integration of container terminal business process and port MIS processes. From the conceptual framework, 4-tier framework shows tightly connected components, with the key elements for data presentation, data process and data interface and middleware (refer Figure 4).

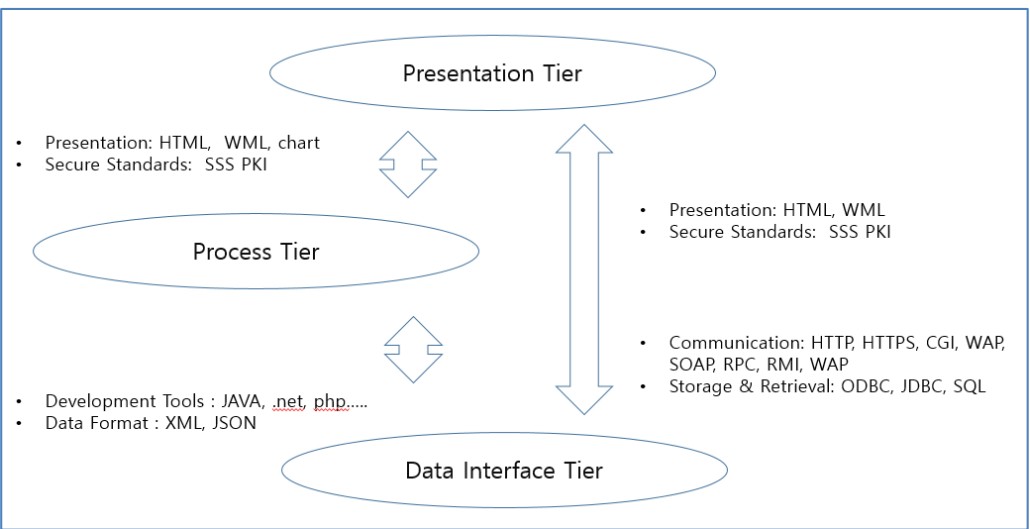

**Figure 4.** Tier of system architecture.

Interface tier establishes a virtual channel over some network infrastructure for any two parties to exchange digital data. This could use a set of message and communication standards to communicate with other users. Key standards for communication include protocols like HTTP (Web-based Hyper Text Transfer Protocol), CGI (Common Gateway Interface), SMTP (Simple Mail Transfer Protocol), and FTP (File Transfer). For message standards, UN/EDIFACT (the United Nations rules for Electronic Data Interchange for Administration, Commerce and Transport), ANSI X12 (American National Standards Institute X12), and Cargo-IMP (Interchange Message Procedures), and file types including Microsoft Excel, Text file, and CSV (Comma Separated Values) file can be used. For secured data transfer, protocols using encryption, such as SSL (Secured Socket Layer) and PKI (Public Key Infrastructure), can be used. Communications can be achieved at application level across the network using RPC (Remote Procedure Call) or RMI (Remote Method Invocation). The Wireless Application Protocol (WAP) is a communication technology that creates an added channel for information exchange using wireless devices, and plays a role in the delivery of MMS (Multimedia Messages Service). For storage and retrieval, DBMS (Database Management Systems) has been used. ODBC (Open Database Connectivity) is a standard interface for accessing a database. Any database that is ODBC-compliant can be accessed using a simple query language, e.g., SQL (Structured Query Language), JDBC (Java Database Connectivity), a special feature for Java, incorporates the functionality of both SQL and ODBC.

Presentation tier specifies how information should be organized when presented, as well as the corresponding presentation format; HTML (Hypertext Markup Language) is a language that formats the information for presentation. XML (eXtensible Markup Language) captures the essence of HTML while adding data structure and data markers to the content, thus providing an information structure for efficient processing and storage. For delivery of content to wireless devices, WML (Wireless Markup Language) specifies a different information structure suitable for wireless usage.

Process tier specifies how information will be retrieved and processed according to the purpose of POI which users want to produce including description, prediction, diagnosis and prescription.

## 5. Results

In this chapter, proof of concept will be attempted with four PPIs. Four areas consist of ship waiting, berth occupancy, yard utilization, and truck turnaround time. To create these PPIs, the ERA diagrams were derived and a database was created based on them. Based on the ERA, we defined an algorithm for generating PPIs.

In the process tier, we will estimate possible indicators in the future by estimating them as time series. In the process tier, we will estimate the indicators that can occur in the future by time series estimation. If problems arise in future indicators, we will find reasons to improve these indicators. The cause of the problem for prescription can be considered in various ways, but the monitoring platform attempted to discover through the interrelationship of the indicators. When a problem-solving method is found, simulation or management science is used to find and implement optimal alternatives. Proof of concept is attempted by comparing the user view designed with the graphic tool and the prototype system output.

### 5.1. PoC of Ship Waiting in Port

The ship's activity at the calling port can be composed of a PFU (Port Facility Usage) entity and SBU (Ship Berthing-Unloading) entity. As the primary keys of SBU and PFU have different attributes, this problem is solved by finding and linking attributes with the same meaning (refer Figure 5). The algorithm for finding ship waiting time is as follows.

· Process 1: In the SBU entity of TOS, a ship is selected based on Terminal Code, Call Sign, and Berthing Time.
· Process 2: Based on Call Sign and Berthing Time, the relevant vessel of PFU entity from PORT-MIS is found.
· Process 3: Determine whether Berthing Time of SBU entity and the Berth Time of PFU entity match.
· Process 4: If there is a difference of more than ±3 h, then go to process 1, or process 5 is performed.
· Process 5: If the facility code of the previous record is the anchorage and the purpose of facility usage is waiting for berth or waiting for cargo, then process 6 is performed
· Process 6: The waiting time can be subtracted the facility use start time from the end time of berth facility. The average ship waiting time is calculated by accumulating individual times and dividing by the number of waiting ships.

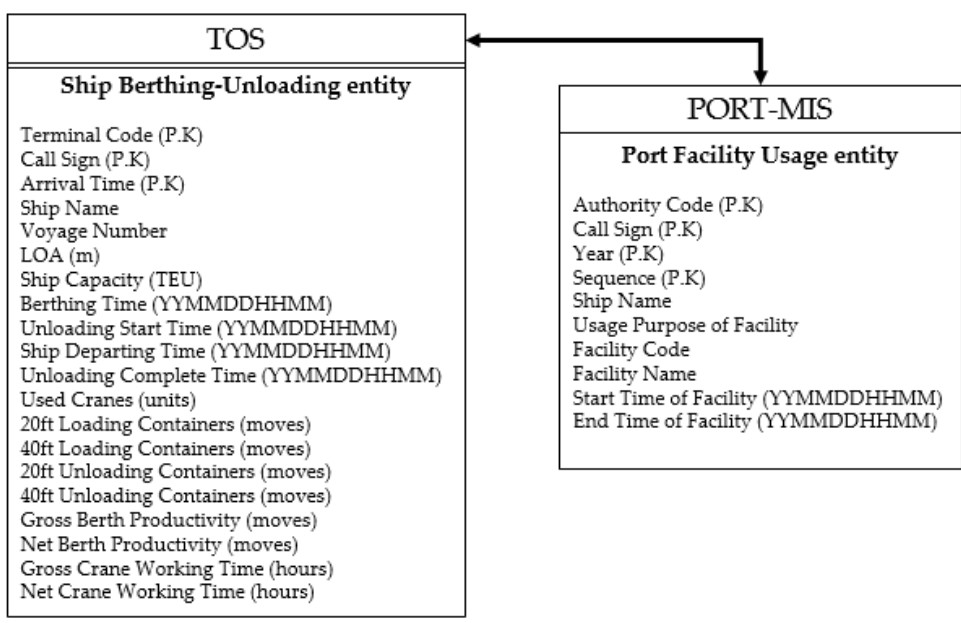

**Figure 5.** ERA diagrams for Ship Waiting PPI.

The ship's waiting is expressed in terms of the number of waiting ships, average waiting time per ship, and ship's waiting rate. The output about the ship's waiting is shown in Figure 6. This shows a trend of 36 months of which the *X*-axis is monthly. The primary *Y* axis on the left shows the waiting time, and the secondary *Y* axis on the right shows number of waiting ships. Referring to Figure 6, in the descriptive perspective [1], the number of ships waiting for the recent month of a terminal is 25, and the average waiting time per ship is 14 min.

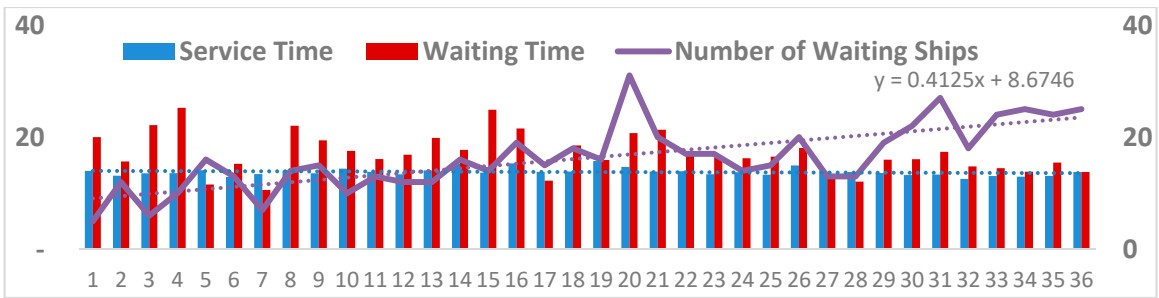

**Figure 6.** User View of Waiting Time, Service Time and Waiting Ships.

Using the time series prediction method for the next 12 months, the number of waiting ships will be 28 per month and the waiting time will be 13 min. It is the number of waiting ships, not the waiting time that matters in the predictions. So, we have to figure out how to reduce the number of waiting ships. Simply diagnose, the number of calling ships and the number of waiting ships correlate 68%, and the service time and the number of waiting ships correlates 65%. If TOC wants to solve this situation, it has to control the number of ships entering the port or to reduce service time in putting more QCs (Quay Cranes).

## 5.2. PoC of Berth Occupancy

The expression Berth Occupancy may also be used and this will be defined as: Berth Occupancy which is formulated by Service Time ÷ Possible Working days in Period (say 363 days per year).

$T_s$(Service Time) is the period of time during which a vessel is berthed in a port whether the ship works or not. The service time will therefore include working and non-working periods [28]. Recently, the GTOs (Global Terminal Operator) method of calculating the Berth Occupancy has evolved [29]. Berth Occupancy is formulated by

$$((\text{LOA}(\text{Length of All}) \times 120\%) \times (T_s + 2\,\text{h})) \div (\text{Length of Berth} \times \text{Possible Working days in Period}).$$

This formula takes into account the time when the berth cannot be used in berthing and leaving the berth and the space of ship's rope on berth.

- · Process 1: In the Berth Facility Specification (BFS) entity of TOS, a terminal is selected based on Terminal Code, and select Length of Berth at the same time (refer Figure 7).
- · Process 2: In the Ship Berthing-Unloading entity of TOS, Call Sign, LOA, Berthing Time, Unloading Complete Time is sequentially selected based on Terminal Code.
- · Process 3: Derive Berth Occupancy using the above equation. Go to Process 2

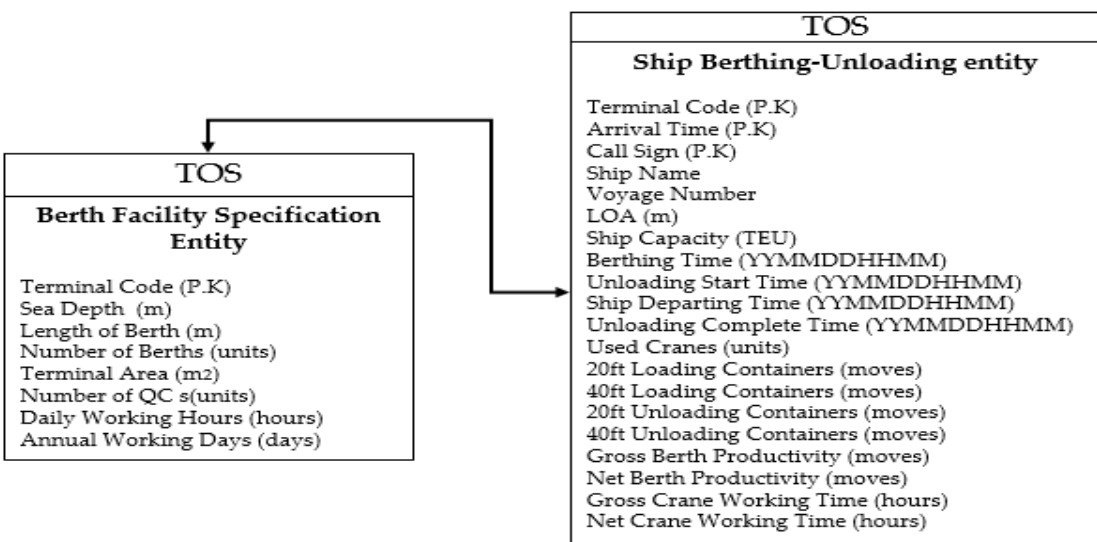

**Figure 7.** Entity Relationship Attribute (ERA) diagram for berth occupancy PPI.

The output about the number of calling ships and berth occupancy is shown in Figure 8. This shows a trend of 36 months of which the *X*-axis is monthly. The primary *Y* axis on the right shows the berth occupancy, and the secondary *Y* axis on the left shows number of calling ships. Referring to Figure 8, in descriptive perspective, the number of calling ships for the recent month of a terminal is 220, and the berth occupancy is 79%. Using the time series method, the number of calling ships after 12 months is predicted 237, and the berth occupancy is predicted 83%. As a result of the forecast, the problem is that the occupancy of the berths increases. This is because, if the berth occupancy increases, there are a lot of waiting ships due to insufficient berth. In order to solve the problem, we have to find which PPI is highly correlated with the occupancy of the berth. It is necessary to check the correlation coefficient, such as the incoming ship, container throughput, or service time.

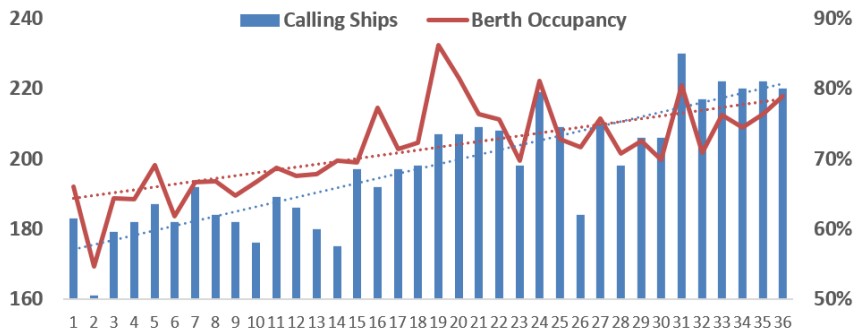

**Figure 8.** User view of calling ships and berth occupancy.

According to correlation analysis, the number of calling ships and berth occupancy correlate 80%. The correlation between container volume and berth occupancy is 87%. Furthermore, the correlation between service time and berth share is 96.8%. Assuming that other factors are out of control, it makes sense to control service time in diagnostic perspective. If terminal A requires control berth occupancy, it has to reduce service time at berth through putting more QCs per ship or increasing QCs productivity as prescription.

### 5.3. PoC of Yard Utilization

Yard-related PPIs include yard utilization, equipment utilization, and container dwell time. Yard utilization is calculated by dividing the container stocked in the yard by the storage capacity. If ODCY (off-dock container yard) is used due to insufficient yard space, this quantity stocked in ODCY is excluded from the stocked quantity. To create a yard-related PPI, entities of yard throughput including container dwell time, yard utilization, equipment utilization by container type are required. The attributes of this entity should be taken from the TOS already calculated. The reason is that the PPI provided by the monitoring platform must be the same as the one calculated by the TOC to have reliability. Here, if there is a difference between the yard utilization calculated by the monitoring platform and the yard utilization received from the TOS, the information must be verified with TOC. The process of calculating yard utilization will be explained using ERA diagram (refer Figure 9).

- Process 1: In Yard Specification entity, the Full Container Storage's Capacity are selected by primary key of Terminal Code and Date. Go to Yard Utilization Entity.
- Process 2: Use the primary key used in Process 1 in the Yard Utilization entity to select the current stocks of the full container.
- Process 3: As Yard Utilization is defined as containers on hand divided by total available slots, use the result of process 1 as the denominator and the result of Process 2 as the numerator to find the yard utilization rate.
- Process 4: Apply Process 2 and 3 to other container types, such as empty, refrigerator, over size, and dangerous container, and calculate the yard utilization rate.

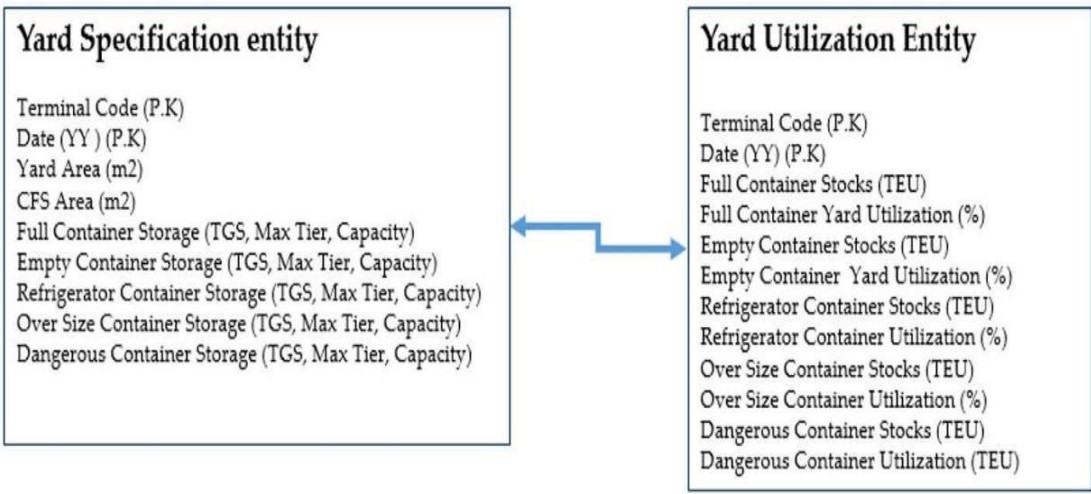

**Figure 9.** ERA diagram for yard-related PPI.

The output about the number of calling ships and yard utilization is shown in Figure 8. This shows a trend of 36 months of which the *X*-axis is monthly and *Y* axis on the left shows yard utilization of full and empty container. Referring to Figure 8, in descriptive perspective, the yard utilization of full and empty containers for the recent month of a terminal is 56% and 81%, respectively. Using the time series method, the yard utilization of full and empty containers after 12 months is predicted at 62% and 87%, respectively.

As a result of the forecast, this problem can be solved by either increasing the empty container storage or removing it with ODCY. However, the first method is limited due to the narrow yard area, and generally the second method is widely used. Another alternative could consider a method of lowering the peak for each day of the week, but it was found that there is a little peak of the yard utilization rate for each day of the week. In particular, the relationship between throughput and yard utilization was found to be a negative relationship (refer Figure 10). It turned out that the adjustment of the quantity of the day of the week did not help improve the yard utilization.

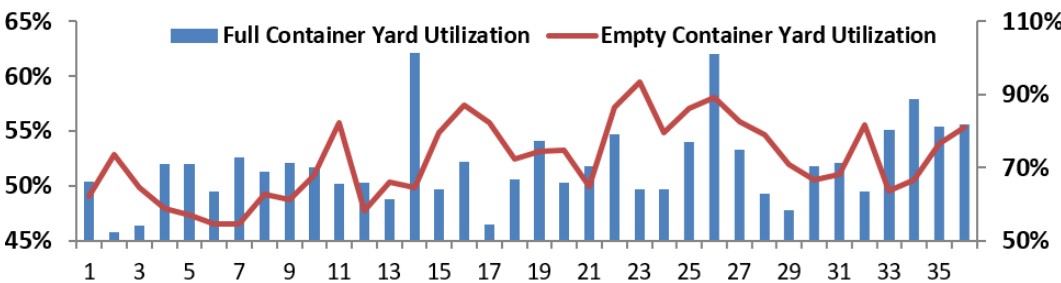

**Figure 10.** User view of full and empty container yard utilization.

### 5.4. PoC of Truck Turnaround Time

Truck turnaround time (TTT) is the total time spent by a truck in the terminal area from gate-in to gate-out for picking and/or dropping a container. It includes the time from the arrival, loading, and unloading of containers, inspecting a truck, completing documentation, and going out from the terminal. TTT is determined by various factors, such as gate in time, travel time from gate in to yard block, working time, and truck driver's meal and break time [29]. The most difficult thing among TTT is that when the truck is loading and unloading at the block, the crane equipment is concentrated on the ship's handling in a result that truck waits in front of the block [4]. The way to reduce TTT is to have enough lanes so that trucks do not wait at the gate and enough space, including parking lot, for trucks to avoid congestion. In addition, when the truck is working on the block, it is to efficiently schedule so that there is no transfer crane waiting.

Truck turnaround time is calculated using CODECO entity. The attributes of CODECO entity are composed of terminal code, truck number, container number, the time of gate-in, the time of gate-out, and in-outbound classification code (refer Figure 11).

The algorithm for finding truck turnaround time is as follows.

- Process 1: In CODECO's entity, the truck number and the time of gate-in are sequentially selected.
- Process 2: Find out whether there is the time of gate-out of the same day with the selected truck number.
- Process 3: If there is the time of gate-out, calculate the difference between the time of gate-in and the time of gate-out.
- Process 4: If there is a difference of more than ±3 h, then go to process 1, or process 5 is performed.
- Process 5 The truck turnaround time can be subtracted the truck gate-in time from the truck gate-out time. The average truck turnaround time is calculated by accumulating individual times and dividing by the number of trucks.

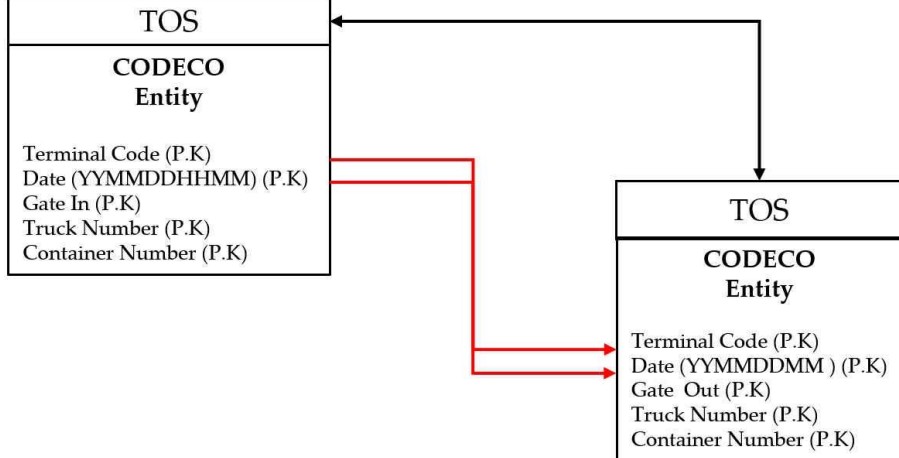

**Figure 11.** ERA diagram for truck turnaround time.

The output about percentage by the truck turnaround time is shown in Figure 12. This shows a trend of 36 months of which the *X*-axis is monthly and *Y* axis on the left shows yard utilization of full and empty container. Referring to Figure 12, in descriptive perspective, if this terminal define the normal truck turn time as 30 min, it takes 27% more than 30 min and 12% more than one hour. The number of trucks carried in the gate per year at this terminal is 270,141, so the number of trucks exceeding 30 min is 72,502 per year. The only problem with truck turnaround time is reducing this time to less than 30 min. In order to reduce the truck turnaround time, it will be a way to lower the yard utilization rate or introduce a take-out reservation system that lowers the peak by time of day.

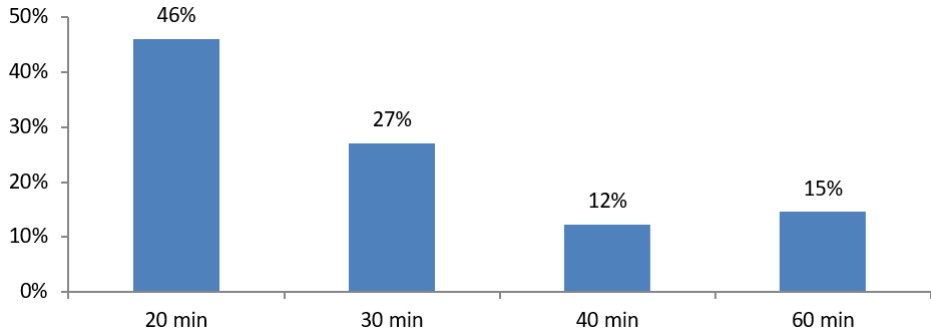

**Figure 12.** User view of the percentage of truck turnaround time.

The correlation between the truck turnaround time and the number of trucks was 95%, which is shown in Figure 13, and the correlation with the yard utilization rate was 50%. If this terminal seeks to solve the problem of longer truck turnaround time, it seems that introducing a truck take-out reservation system is more effective than lowering the yard utilization rate.

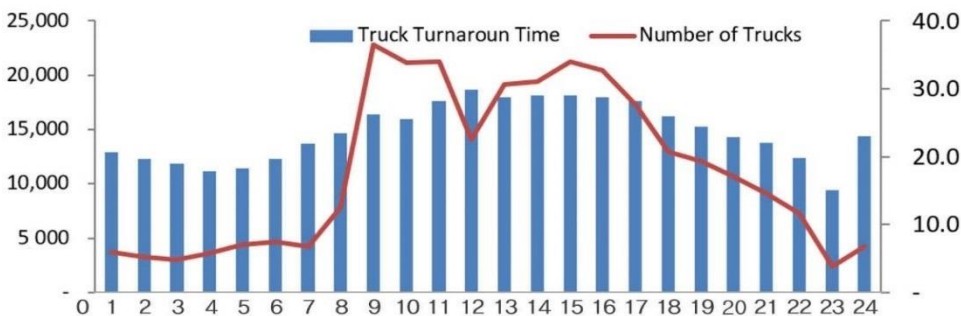

**Figure 13.** User view of the number of trucks and truck turnaround time.

## 6. Conclusions

Along with the proliferation of the 4th industrial revolution, there is a situation in which the port stakeholders grasp its performance in real time and respond appropriately. This study is to prove whether the PPI platform system can be developed to solve problems in the field. As a research tool necessary for proof of concept, a prototype of the PPI monitoring platform was developed.

The prototype consists of a data interface tier, a presentation tier, a process tier and a middleware tier. In the data interface tier, the development of a demon to automatically collect the data of TOS of an external entity, standardization of exchange data, such as XML, and standardization of the database structure of TOS are included. In the presentation tier, PPI is expressed in time series using graphic tools. When searching for a desired PPI, the terminal name and a certain period are entered.

In this paper, proof of concept was attempted with four PPIs. Four areas were targeted: ship waiting, berth occupancy, yard utilization, and truck turnaround time. To create these PPIs, the ERA diagrams

were derived and a database was created based on them. Based on the created database, we defined an algorithm for generating PPIs and user view through graphic tool of Excel spread sheet. Proof of concept (PoC) was attempted by comparing the user view designed with the graphic tool and the prototype system output.

The research result is a PoC study to check whether the service level of the container terminal can be monitored using an information platform. The study has the limitation to implement the prototype system, which is a difficulty in the provision of real-time service levels. In other words there is a feeling of being late to immediately grasp the efficiency of operation and improvement of productivity of port resources by providing an indicator of activity that has already passed. In order for the monitoring system to operate normally, data collection must be performed in real time, but legal or institutional regulation on data submission are currently insufficient. Therefore, it is judged that the regulation must be established in advance.

In the future, it is necessary to make a plan, make regulation for collecting real time data from TOC, design an organization, and establish budget for developing commercial system in order to operate the monitoring platform practically.

**Author Contributions:** N.K.P. contributed to the overall idea, data collecting, analyzing and writing of this study. J.H.L. contributed the editing and figures drawings of this study. All authors read and approved this paper. All authors have read and agreed to the published version of the manuscript.

**Funding:** This research received no external funding.

**Conflicts of Interest:** The authors declare no conflict of interest.

## Appendix A

**Table A1.** Terminal used for data collection.

| No | Port | TOC |
|---|---|---|
| 1 | | Hutchison Korea (HBCT) |
| 2 | | Busan Port Terminal (BPT) |
| 3 | | Dongbu Busan Container Terminal (DPCT) |
| 4 | | Intergis Co., Ltd. |
| 5 | Busan Port | Busan New Port International Terminal (PNIT) |
| 6 | | Busan New Port (PNC) |
| 7 | | Hanjin Busan Container Terminal (HBCT) |
| 8 | | Hyundai Busan New Port (HPNT) |
| 9 | | Busan New Container Terminal (BNCT) |
| 10 | | Incheon International Container Terminal (ICT) |
| 11 | Incheon | Sunkwang |
| 12 | | Hanjin Incheon Container Terminal (HICT) |
| 13 | | E1 Container Terminal (E1) |
| 14 | Gwangyang | SM Gwangyang Terminal (SM) |
| 15 | | Korea International Terminal (KIT) |
| 16 | Ulsan | Dongbang Container Terminal (DCT) |
| 17 | | Jeongil Ulsan Container Terminal (JUCT) |
| 18 | Pyeongtaek | Pyeongtaek Container Terminal (PCT) |
| 19 | | Pyeongtaek Dongbang Iport (Iport) |

The data about berth facility consist of QC (Quay Crane), working time, annual throughput, and terminal area to be collected in Table A1.

**Table A2.** Berth facility data to be collected.

| Berth | | | QC | | Working Time | | Annual Throughput | Terminal Area (M2) |
|---|---|---|---|---|---|---|---|---|
| Depth of Sea | Length of Berth (m) | Number of Berth | Type | Number | Daily (hours) | Annual (days) | | |

The data about storage facility consist of TGS (Twenty Feet Ground Slots), maximum tier, and stocking capacity in Table A2.

**Table A3.** Storage facility data to be collected.

| Storage Area (m². ) | Full Storage | | | Empty Storage | | | Refrigerate Storage | | | Dangerous Storage | | | Over Size Storage | | |
|---|---|---|---|---|---|---|---|---|---|---|---|---|---|---|---|
| | TGS | Tier | Cap | TGS | Tier | Cap | TGS | Tier | Cap | TGS | Tier | Cap | TGS | Tier | Cap |

The data about berth activity consist of ship name, call sign, ship capacity, ship arrival time, ship berth time, number of containers to be loaded and unloaded, QC working hours, GBP (Growth Berth Productivity), and NBP (Net Berth Productivity) in Table A3.

**Table A4.** Berth activity to be collected.

| Ship Name | Call Sign | Ship Capacity (TEU) | Voyage Number | Ship Arrival Time | Ship Berthing Time | Loading Container (moves) | Unloading Container (moves) | QC Work Hours | GBP/NBP (moves) |
|---|---|---|---|---|---|---|---|---|---|

The data about storage activity consist of daily date, number of container stocked in full storage and its occupancy, number of container stocked in empty storage and its occupancy, number of container stocked in refrigerator storage and its occupancy, and number of container stocked in dangerous storage and its occupancy in Table A4.

**Table A5.** Storage activity data to be collected.

| Daily date | Full Container (TEU) | | Empty Container (TEU) | | Refrigerator Container (TEU) | | Dangerous Container (TEU) | |
|---|---|---|---|---|---|---|---|---|
| | Number of Stocking | Stocking Occupancy (%) | Number of Stocking | Stocking Occupancy (%) | Number of Stocking | Stocking Occupancy (%) | Number of Stocking | Stocking Occupancy (%) |

The data about truck activity at yard consist of truck number, the type of gate in or gate out, the number of container, the type of laden or empty of truck, the time of containers gate-in or gate-out, and the time of truck gate-in or gate-out in Table A5.

**Table A6.** Truck turnaround time data items and format to be collected.

| Truck Number | Gate in or Gate Out | Number of Containers | Container Size | Laden or Empty on Truck | Container | | Truck | |
|---|---|---|---|---|---|---|---|---|
| | | | | | 1st container | 2nd container | Gate-In Time | Gate-Out Time |

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
