# Peer review of "PoC (Proof of Concept) for Performance Monitoring Platform of Container Terminals"

_jmse, doi:10.3390/jmse8120971_

Round 1

Reviewer 1 Report

This study focuses on the proof of concept for performance monitoring platform of container terminals. I think the paper fits well the scope of the journal and addresses an important subject. However, a number of revisions are required before the paper can be considered for publication. There are some weak points that have to be strengthened. Below please find more specific comments:

*Please spell out “PoC” in the title. Some readers may not be able to understand the abbreviation from the first glance.

*Page 1 line 12: There is no need to capitalize “government”.

*The abstract seems to be somewhat short. Please try to expand the abstract a bit and provide more information regarding the contributions of this work along with important outcomes.

*It seems that the spacing between words is often missing (e.g., “PoC(Proof of Concept)” should be replaced with “PoC (Proof of Concept)”, “Port Authority(PA)” should be replaced with “Port Authority (PA)”, etc.). Please check the entire manuscript and address this issue.

*Page 2: The authors start section 2.1 with the discussion regarding the berth availability and berth scheduling. I recommend expanding this discussion by highlighting the importance of effective berth scheduling on the overall performance of marine container terminals. This discussion should be supported by the recent and relevant references related to berth scheduling, including the following:

Luo, C., Fei, H., Sailike, D., Xu, T. and Huang, F., 2020. Optimization of Continuous Berth Scheduling by Taking into Account Double-Line Ship Mooring. Scientific Programming, 2020.

Dulebenets, M.A., 2020. An Adaptive Island Evolutionary Algorithm for the berth scheduling problem. Memetic Computing, 12(1), pp.51-72.

Malekahmadi, A., Alinaghian, M., Hejazi, S.R. and Saidipour, M.A.A., 2020. Integrated continuous berth allocation and quay crane assignment and scheduling problem with time-dependent physical constraints in container terminals. Computers & Industrial Engineering, 147, p.106672.

Wawrzyniak, J., Drozdowski, M. and Sanlaville, É., 2020. Selecting algorithms for large berth allocation problems. European Journal of Operational Research, 283(3), pp.844-862.

Kavoosi, M., Dulebenets, M.A., Abioye, O.F., Pasha, J., Wang, H. and Chi, H., 2019. An augmented self-adaptive parameter control in evolutionary computation: A case study for the berth scheduling problem. Advanced Engineering Informatics, 42, p.100972.

Tasoglu, G. and Yildiz, G., 2019. Simulated annealing based simulation optimization method for solving integrated berth allocation and quay crane scheduling problems. Simulation Modelling Practice and Theory, 97, p.101948.

*Page 5 line 143: “CODECO information collected 56,198,183 cases” does not sound well. Do you mean “CODECO information included 56,198,183 cases”.

*Page 8: Again, please make sure that there is spacing between words. I see a lot of merged words in Fig. 5.

*Page 11: My previous comment applies to Fig. 11 and other relevant figures as well.

*Page 12: The author show truck turnaround times. It would be good to add a discussion on different approaches that can be used to reduce the total truck turnaround times.

*Page 13: Moreover, the conclusions section should expand more on limitations of this study and future research needs.

Author Response

Please, refer attathed file. Thanks.

Reviewer 2 Report

The paper represents a case study by introducing a concept of a performance monitoring system for container ports.

In general the content is of interest for the reader and the system itself is described properly.

Thertheless there is some room for improvement

1. Research Methodology

The paper needs a proper description of the methodology applied. At least the requirements for such a system shall be derived and introduced and the findings shall critically reflect the requirements

2. Description of the Paper structure / introductions

The paper reads like a list of topics. Please provide an argumented overview of the paper structure in the beginning and provide introductory statements to the chapter so the reader can orientate him/herself.

3. Describe related work

Desicribe related research, systems etc. and explain why this paper provides a contribution to science / knowledge

Author Response

Please, refer the attatched file. Thanks.

Round 2

Reviewer 1 Report

The authors took seriously my previous comments and made the required revisions in the manuscript. The quality and presentation of the manuscript have been improved. Therefore, I recommend acceptance.